# Combination of Conventional Drugs with Biocompounds Derived from Cinnamic Acid: A Promising Option for Breast Cancer Therapy

**DOI:** 10.3390/biomedicines11020275

**Published:** 2023-01-19

**Authors:** Lyvia Eloiza de Freitas Meirelles, Maria Vitória Felipe de Souza, Lucimara Rodrigues Carobeli, Fabrício Morelli, Natália Lourenço Mari, Edilson Damke, Cristiane Suemi Shinobu Mesquita, Jorge Juarez Vieira Teixeira, Marcia Edilaine Lopes Consolaro, Vânia Ramos Sela da Silva

**Affiliations:** Department of Clinical Analysis and Biomedicine, State University of Maringá, Maringá 87020-900, PR, Brazil

**Keywords:** breast neoplasms, therapeutics, drug combinations, cinnamates, drug synergism

## Abstract

Despite the options available for breast cancer (BC) therapy, several adverse effects and resistance limit the success of the treatment. Furthermore, the use of a single drug is associated with a high failure rate. We investigated through a systematic review the in vitro effects of the combination between conventional drugs and bioactive compounds derived from cinnamic acid in BC treatment. The information was acquired from the following databases: PubMed, Web of Science, Embase, Scopus, Lilacs and Cochrane library. We focused on “Cinnamates”, “Drug Combinations” and “Breast neoplasms” for publications dating between January 2012 and December 2022, based on the PRISMA statement. The references of the articles were carefully reviewed. Finally, nine eligible studies were included. The majority of these studies were performed using MCF-7, MDA-MB-231, MDA-MB-468 and BT-20 cell lines and the combination between cisplatin, paclitaxel, doxorubicin, tamoxifen, dactolisib and veliparib, with caffeic acid phenethyl ester, eugenol, 3-caffeoylquinic acid, salvianolic acid A, ferulic acid, caffeic acid, rosmarinic acid and ursolic acid. The combination improved overall conventional drug effects, with increased cytotoxicity, antimigratory effect and reversing resistance. Combining conventional drugs with bioactive compounds derived from cinnamic acid could emerge as a privileged scaffold for establishing new treatment options for different BC types.

## 1. Introduction

Cancer is a complex disorder caused by the uncontrolled growth of abnormal cells that may affect several parts of the body. Worldwide, breast cancer (BC) is still the most frequent malignancy, representing 11.7% of all cancer cases in this population, with an estimated 2.3 million new cases and 684,996 deaths in 2020. The BC burden is expected to be more than 3 million new cases in 2040, meaning a 33.8% increase from 2020 [1]. BC is a heterogeneous disease manifesting diversity at molecular, histological and clinical levels [2]. This cancer is multifactorial and involves genetic predisposition (5–10%) [3], lifestyle and other environmental factors [4]. It has been estimated that approximately 20% of BC can be attributed to modifiable risk factors [5]. Normally, BC starts with ductal hyperproliferation and later evolves into benign tumors or carcinomas that may be metastatic due to several carcinogenic factors [6]. BCs can be stratified into different entities based on clinical behavior, histologic features and/or biological properties, which provide the basis for deciding on standard treatment and planning for follow-up [7,8,9].

Concerning the tissue that the BC is originated from, there are two main types in the breast: the milk or mammary ducts and the breast lobules which determine the clinical progression and behavior of the disease [10]. Cancers originating from milk ducts, classified as ductal carcinoma, account for 40–75% of all diagnosed BCs. In contrast, cancers originating from breast lobules, classified as lobular carcinoma, are less prevalent, accounting for ~10–15% of diagnosed BCs [11]. Lobular carcinomas are more likely to overexpress hormone receptors such as estrogen (ER) or progesterone (PR) [12,13]. By histopathologic analysis, they are classified as ductal carcinoma in situ, invasive ductal carcinoma, lobular carcinoma in situ, invasive lobular carcinoma, inflammatory carcinoma and other less common types [14] (Figure 1).

The molecular stratification of BCs is primarily based on gene expression profiling; this also includes the expression status of hormonal receptors, such as the ER and PR, as well as human epidermal growth factor receptor 2 (HER2) which is also known as ERBB2. Based on this, BCs are classified into four main subtypes (regardless of tissue origin): luminal ER-positive (luminal A and luminal B), HER2 enriched and triple negative receptors (basal-like) (TNBC) [15]. More specifically, luminal A is ER/PR positive and HER2 negative, luminal B is ER/PR positive and HER2 expression is variable, HER2-overexpression is ER/PR normally negative and HER2 positive, and TNBC is ER/PR negative and HER2 negative [7,16] (Figure 1). These classifications also help to determine the prognosis of the disease and the treatment to be implemented [17]. In general, ER/PR+ BC present the best prognosis (70% of all cases), followed by HER2+ cancers (~15–20% of all cases) and TNBC (~10%) which has the worst clinical prognosis and survival rate [18]. 

Overall, the therapy against BCs involves a multimodal strategy with a combination of surgery for operable tumors, radiotherapy, chemotherapy, targeted therapy, immunotherapy and hormone therapy [19]. In clinical practice, chemotherapy has been the core treatment strategy for metastatic BC for several decades and remains a crucial component of treatment regimens. Chemotherapeutic agents are appropriate for most patients with metastatic tumors, including those with hormone receptor-positive (HR+) disease with extensive visceral involvement, HR+ disease after failure of hormone-directed therapy, HER2+ disease and TNBC [19]. The main drugs approved and currently used for BC chemotherapy are capecitabine, cyclophosphamide, docetaxel, doxorubicin hydrochloride, epirubicin hydrochloride, 5-fluorouracil, gemcitabine hydrochloride, paclitaxel and trexall [4]. Still, in terms of current treatments, hormone therapy is the standard of care for ER+ and/or PR+ BC patients [20]. Some options of drugs include tamoxifen and toremifene, two selective ER modulators approved for metastatic BC; and the antiestrogen fulvestrant for postmenopausal women with metastatic ER+ BC [21]. These treatments are innovative and important to improve the response and survival to BCs, but are associated with several adverse effects and may develop drug resistance within a few months [20,22]. Even more, BC treatment using a single drug is associated with a high failure rate due, in part, to the heterogeneity of drug response within individuals, nonspecific target action, drug toxicity and/or development of resistance [23]. In this sense, the combined therapies against cancer are indicated. In comparison to monotherapy, polytherapy that simultaneously targets distinct mechanisms is less likely to fail, particularly when using non-cross-resistant treatments [24]. Despite being promising, there are several challenges to the use of combined therapies, such as limited drug options or access to them and higher treatment-related toxicity when using combinations of chemotherapies [24]. So, investigating new combinations for combating resistance and improving cancer treatment is currently of great interest in the clinical setting and in computational modeling [22].

In order to evaluate the effectiveness of new drug combinations, there is a continuous need to search new compounds for BC treatment. In this sense, parallel to the development of synthetic drugs, substantial attention has focused on natural products with anticancer properties, which has stimulated the search for therapeutic alternatives [25]. Phenolic acids have a carboxyl functional group and are divided into two classes: hydroxybenzoic acids and hydroxycinnamic acids [26]. They are characterized by having a benzene ring, a carboxylic group and one or more hydroxyl and/or methoxyl groups in the molecule, providing antioxidant properties, being, therefore, indicated for the treatment and prevention of cancer, cardiovascular disease and other diseases [27,28]. More specifically, hydroxycinnamic acids are present in various foods and beverages of plant origin, such as coffee, yerba mate, apple, plum and other fruits, crucifers and cereals, among others [29]. Examples of this class of compounds are caffeic, p-coumaric, ferulic and sinapic acid which, in most foods, are esterified to quinic acid, tartaric acid or carbohydrates and derivatives [30,31,32]. As they are widely available and present potential anticancer effects, the derivatives of cinnamic acid have been researched as antitumor agents [33,34].

The search for combinatorial treatments has increased significantly in recent years to achieve greater biocompatibility, less toxicity and better therapeutic potential, which can simultaneously target many of the differential weaknesses of the cancer. Therefore, this review aims to briefly discuss the in vitro studies that evaluated the combination of conventional drugs with bioactive compounds derived from cinnamic acid in BC treatment, which may contribute to the development of new therapeutic options. 

## 2. Materials and Methods

This systematic review was conducted according to the Preferred Reporting Items for Systematic Reviews and Meta-Analyses (PRISMA) Statement 2020 [35] focusing on the use of combination drugs with bioactive compounds derived from cinnamic acid in BC treatment.

### 2.1. Search Strategies and Inclusion Criteria

Relevant studies of literature were first identified through electronic searches. A comprehensive and systematic search of relevant databases (PubMed, Embase, Web of Science, SCOPUS, Lilacs and Cochrane library) was conducted to identify original studies published in peer-reviewed journals in English in the last 10 years (1 January 2012 and 31 December 2022), which explored the use of combination drugs with bioactive compounds derived from cinnamic acid against BC.

The definition of MeSH terms was carried out in-depth and by consensus by the six researchers of group 1 (L.E.d.F.M, M.V.F.d.S, L.R.C, F.M, N.L.M and C.S.S.M). Nearby, the researcher conducted a literature search by six researchers from group 1 using medical subject heading (MeSH) terms and a combination of several keywords (e.g., “Breast neoplasms” (MeSH) AND “Cinnamates” (MeSH) OR “Phenolic acid” (MeSH) OR “Phenylpropionates” (MeSH) AND “Drug Synergism” (MeSH) OR “Drug Therapy, Combination” (MeSH) OR “Drug Combinations” (MeSH). Databases and search strategy used, and numbers of retrieved studies in Appendix A.

### 2.2. Study Selection

Titles and abstracts were carefully selected to ensure publication originality, quantitative and qualitative consensus by group 1. The studies initially selected had to fit the following two criteria: the first criterion included original and clinical studies involving a combination of drugs with bioactive compounds derived from cinnamic acid in BC treatment. The second criterion was to exclude duplicate studies, review studies, letters to the editor, comments, abstract congress and books. After consensus, the papers most closely associated with the theme descriptors were selected. Then, the full-text articles were randomly distributed among the investigators of group 1 who acted as independent evaluators in charge of the inclusion of articles in the final cohort. 

### 2.3. Data Extraction and Analysis

The reviewers of group 1A (L.E.d.F.M, M.V.F.d.S, L.R.C, F.M and C.S.S.M) used a standardized data abstraction form to capture information on authors, year of publication, BC cell lines, drugs and compounds used, methods and main results found. Differences in data abstraction were resolved by consensus. Data were analyzed using Excel^TM^ to display all relevant information in an organized manner. To increase the sensitivity of the search, the references of the original articles were carefully reviewed for recovery articles that could be additionally used in this review. To ensure that all relevant data from each paper were included in the review, a final consensus was achieved following an additional examination of the full texts by two individual experts, group 2 (M.E.L.C and V.R.S.d.S).

### 2.4. Risk of Bias Assessment

The quality and risk of bias in the selected papers were performed independently by three researcher specialists (L.E.d.F.M, M.V.F.d.S and F.M) based on the CONSORT (Consolidated Standards of Reporting Trials) guidelines [36]. For the analysis of the 9 in vitro studies, we used a checklist composed of 14 domains (structured abstract; scientific background and rationale; objectives and/or hypotheses; intervention of each group; outcomes; sample size; randomization: sequence generation; allocation concealment mechanism; implementation; blinding; statistical methods; outcomes and estimation; limitations; funding; protocol). These domains are assigned in (+) Low risk of bias; (-) High risk of bias; (?) Unclear risk of bias, and the results are available in Appendix A.

## 3. Results and Discussion

A total of 49 articles from PubMed, Embase, Web of Science, SCOPUS, Lilacs and Cochrane library were found. Four articles were excluded due to duplicates, 11 were ineligible (reviews and outside of 10 years) and further analysis of the titles and abstracts of the remaining 34 articles led to the exclusion of 20 articles (excluded based on title and abstract). In addition, 6 articles were excluded after full reading and 1 article was retrieved by previous search. Finally, 9 articles with in vitro studies that evaluated the combination of conventional drugs with bioactive compounds derived from cinnamic acid in BC treatment were included in the analysis. The flow chart of the article selection process is presented in Figure 2 and the studies included in the systematic review are in Appendix A.

In general, of the 9 studies included, 5 were performed in a single ER+ cell line, 4 in MCF-7 [37,38,39,40] and 1 in MCF-7/PTX (MCF-7 paclitaxel-resistant cell line) [41]; 1 was performed only in the MDA-MB-231 cell line [42]; 1 in three different cell lines (MDA-MB-231, MDA-MB-468, and BT-20) [43], which originates from TNBC; 1 in MCF-7 (ER+) and MDA-MB-231 (TNBC) cell lines [44]; and 1 in different patient tumor samples, more specifically by the ex vivo organotypic culture of human BC explants [45]. Regarding the drugs used for the combined treatment with cinnamic acid biocompounds, most were chemotherapy, mainly with cisplatin and paclitaxel (*n* = 2, for each), followed by doxorubicin (*n* = 1) [37,40,41,43,45]. Other drugs used were hormone therapy with tamoxifen (*n* = 2) [38,39] and targeted therapy with drugs in the advanced clinical study phase (*n* = 2) named NVP-BEZ-235 (dactolisib) and ABT-888 (veliparib), phase II and III study drugs, respectively [42,44]. Finally, 8 different biocompounds from cinnamic acid were evaluated, mainly the caffeic acid phenethyl ester (CAPE; *n* = 3) [38,39,42], followed by eugenol (*n* = 2) [40,43], 3-caffeoylquinic acid [37], salvianolic acid A [41], ferulic acid [44] and caffeic acid, rosmarinic acid and ursolic acid in the same study [45] (*n* = 1 for each). Table 1 summarizes the data from the 9 included studies.

### 3.1. Highlights of Studies Performed in MCF-7 Cell Line (ER+)

The five studies included generally showed excellent results between the combination of conventional drugs (cisplatin, tamoxifen, doxorubicin and paclitaxel) with cinnamic acid bio derivatives (3-caffeoylquinic acid, CAPE, eugenol and salvianolic acid A) in the MCF-7 cell lines that represents BC ER+ [37,38,39,40,41]. More specifically, Suberu et al. [37] investigated the cytotoxicity of 3-caffeoylquinic acid, a caffeic derivative that is one of the major constituents of artemisia tea, in combination with cisplatin. The authors showed that the 50% inhibitory concentration (IC_50_) of cisplatin was 2.5-fold lower when in combination with 3-caffeoylquinic acid compared to cisplatin alone [37].

Motawi et al. [38] evaluated the combined effects of tamoxifen and CAPE. The authors showed synergistic cytotoxic effects, manifested by significant activation of the apoptotic machinery, along with downregulation of protein levels of B-cell lymphoma-2 (Bcl-2) and beclin-1, upon using the combination regimen. Moreover, a decrease in vascular endothelial growth factor (VEGF) level was detected. These results suggest that CAPE relatively improved the anticancer activity of tamoxifen via its apoptotic and angiostatic potentials [38]. In another study, the same group performed the efficacy of tamoxifen and CAPE combination on multiple targets. An increase in caspase-3 activity, apoptosis rates, glutathione level and nitric oxide production was observed. Gene expression of Bcl-2, beclin 1 and VEGF was reduced in a time-dependent manner (24 and 48 h) in all treatment regimens, with the combination showing the most potent effect. However, no significant change in tamoxifen uptake was observed with CAPE combined treatment [39]. 

The studies described above presented the ability of 3-caffeoylquinic acid and CAPE to reduce the effective dose of cisplatin and tamoxifen, which provides a rationale for future experimental and clinical investigations of this combination for HR+ BC treatment. Among natural products derived from cinnamic acid with anticancer properties, the potential of caffeic acid and its naturally derived CAPE has been previously evaluated, which has shown activity when used alone or synergistically with other antitumor agents, being able to induce cell cycle arrest and apoptosis in tumors [33]. Furthermore, caffeic acid, as well as caffeine, have already been associated with the suppression of mammary carcinogenesis in vivo. However, the association between coffee intake and BC control has not yet been confirmed [46]. 

Fouad et al. [40] tested the effect of combining drugs with other cinnamic acid biocompounds on MCF-7 cells. They investigated whether the epigenetic and immunomodulatory effects of eugenol (4-allyl-2-methoxyphenol) could enhance doxorubicin cytotoxicity and observed a significant increase in doxorubicin cytotoxicity and a synergistic cytotoxic effect by eugenol and doxorubicin. Furthermore, the combination of compounds resulted in 1: a fivefold increase in the percentage of cells in G0/G1 and induced apoptosis through the higher BAX/Bcl-2 ratio; 2: decreased protein expression of luminal differentiation marker cytokeratin 7 (CK7), which is associated with resistance to doxorubicin treatment; 3: reduced protein expression of microtubule associated protein 1 light chain 3 beta II (LC3BII), which is important because the main reason for the acquired resistance phenotype in ER+ BCs and its molecular target LC3B is found to be highly expressed in the BC tissues; 4: induced level of global histones acetylation along with increasing the protein expression of histone acetyltransferase, contributing to its proautophagic effect and intrinsic apoptotic cell death [40]. Previous evidence suggests that eugenol can affect cancer cells as an antioxidant, preventing mutation and as a pro-oxidant, influencing signal pathways and killing cancer cells [47]. Its anticancer effects are accomplished by various mechanisms such as inducing cell death, cell cycle arrest, inhibition of migration, metastasis and angiogenesis on several cancer types, such as leukemia, lung cancer, colon, colorectal, skin, gastric, breast, cervical and prostate, whereas it has insignificant toxicity towards normal cells [48,49]. In addition, doxorubicin is a chemotherapeutic commonly used in the treatment of HR+ BC patients with poor prognostic features, but unfortunately, the optimal clinical use of this drug is usually limited to the development of multidrug resistance [50,51]. In this sense, the results of this study were very promising as they showed that eugenol enhanced the cytotoxic activity of doxorubicin through an apoptotic approach, indicating synergism to a doxorubicin–cytotoxic effect by eugenol on HR+ BC cells [40]. 

Moreover, Zheng et al. [41] evaluated the effects of salvianolic acid A and paclitaxel on the resistance, migration and invasion of MCF-7/PTX cells. The authors observed that treatment could reverse the MCF-7/PTX cells resistance to paclitaxel and markedly inhibit tumor migration and invasion. The importance of these results is evidenced by the fact that paclitaxel is another drug widely applied in first-line chemotherapy for treating BC and can present resistance. Furthermore, transgelin 2, an actin cross-linking/gelling protein, has been associated with an oncogenic role in the development of human tumors. Additionally, there is some evidence that transgelin 2 is associated with resistance to chemotherapeutic treatments and with tumor migration and invasion [52,53,54]. These results highlight that combined salvianolic acid A and paclitaxel treatment could reverse this resistance, inhibit the migration and invasion and suppress the expression of transgelin 2, which could be useful in ER+ BC treatments [41].

The literature describes that the favorable outcomes for synergism may include the increasing in the therapeutic efficacy; decreasing the dosage with increasing or maintenance of the same efficacy avoiding toxicity; minimizing or slowing down the development of drug resistance; and providing selective synergism against the target (or efficacy synergism) versus host (or toxicity antagonism) [22]. Collectively, these studies suggested that the combined therapy of conventional drugs with cinnamic acid bio derivatives appears to be very promising as a future therapeutic option for BC ER+, including those resistant to paclitaxel. Therefore, additional preclinical in vitro and in vivo studies with 3-caffeoylquinic acid, CAPE, eugenol and salvianolic acid A as well as new bio derivatives of the acid need to be stimulated.

### 3.2. Highlights of Studies Performed on TNBC Cell Lines

Three studies were performed using TNBC cell lines (MDA-MB-231, MDA-MB-468 and/or BT-20 cells), which represent the most aggressive and worst prognostic type of BC. Recalling, TNBC is HER2-, ER- and PR- [55,56]. Because of these characteristics, they are not sensitive to endocrine therapy or trastuzumab and chemotherapy is the main systemic medical treatment [55]. Moreover, TNBC presents high metastatic potential and higher chances of poor recurrence when compared to other BC subtypes [57,58]. The search for new treatment options that can address the characteristics of this tumor subtype is necessary.

Of these three studies in TNBC cell lines, two were performed using the combination of biocompounds derived from cinnamic acid with drugs in phase II (dactolisib) and III (veliparib) of the clinical study [42,44]. Choi et al. [44] evaluated the effect of combined therapy of poly (ADP-ribose) polymerase (PARP) inhibitor, veliparib, with ferulic acid in MCF-7 and MDA-MB-231 cell lines. They demonstrated that ferulic acid sensitizes BC cells to veliparib and reduces colony formation. These results indicate that this combination could be a potential chemotherapeutic strategy for ER+ BC and TNBC [44]. 

Torki et al. [42] evaluated the effect of combined treatment of dactolisib plus CAPE on the expression of some transcription factors (forkhead box O3, pFOXO3), signaling proteins (Protein kinase B, AKT), receptors (C-X-C chemokine receptor type 4, CXCR4, a marker for metastasis), as well as apoptosis and cell growth in MDA-MB-231 cells, in the presence of transforming growth factor-β1 (TGF-β1) as a cytokine in the cancer microenvironment. They observed the inhibition of cell viability, growth, reduction of marker for metastasis (CXCR4) expression and decrease in the expression of p-FOXO3a in a time-dependent manner. The authors suggested that tumor metastasis and progression in MDA-MB-231 cells can be effectively reduced through the combined use of dactolisib and CAPE [42]. This hypothesis is supported by the following evidence. The tumor microenvironment in TNBC is rich in TGF-β1 cytokine, which activates several intracellular signaling pathways such as phosphatidylinositol 3-kinase/AKT/mammalian target of rapamycin (PI3K/Akt/mTOR) and induces cell proliferation, cell differentiation, tumor progression and metastasis [59,60]. FOXO3a is a downstream target of the (PI3K)/Akt pathway and is known to be a prognosis marker for BC [61,62,63,64]. Previous studies have shown that the treatment with CAPE in different cancer cells inhibited NF/κB activation and induced apoptosis via activation of caspases and down-regulation of anti-apoptotic proteins [65,66]. Additionally, CAPE induced cell cycle arrest by the suppression of cyclin proteins (both D and E type) and c-Myc expression [67,68]. Moreover, dactolisib is a dual reversible PI3K/mTOR inhibitor, which completely inhibits both normal and mutant PI3K and mTOR [69]. A significant decrease in tumor growth has been shown in dactolisib-treated tumors [70]. 

Finally, Islam et al. [43] investigated the capacity of eugenol in enhancing the chemotherapeutic potential of cisplatin in cells representing TNBC (MDA-MB-231, MDA-MB-468 and BT-20). The authors showed that eugenol enhanced the chemotherapeutic potential of cisplatin against TNBC cells and enhanced the inhibition of aldehyde dehydrogenase (ALDH) enzyme activity and the nuclear factor kappa B (NF-κB) signaling pathway [43]. ALDH expression is an independent prognostic factor for BC patients, including the TNBC subtype [71,72]. Active NF-κB has been identified as an important mechanism of cisplatin resistance in different tumor cells [73]. Furthermore, other studies have shown that eugenol combined with chemotherapy may lead to a boosted effectiveness with decreased toxicity and adverse effects of chemotherapy agents such as gemcitabine and cisplatin [43,74,75].

The results of these three studies carried out in TNBC cell lines are very promising as new treatment options for TNBC, the most aggressive and worst prognostic type of BC [55,56]. These data reinforce the importance of the results obtained by these three studies. The combined treatment of conventional drugs with bioactive compounds derived from cinnamic acid present great potential and, therefore, the performance of new tests in vitro and in vivo with ferulic acid, CAPE, eugenol and others not yet tested, need to be encouraged.

### 3.3. Highlights of the Study Performed in BC Patient’s Tumor Samples

Carranza-Torres et al. [45] tested a combined therapy of paclitaxel with caffeic acid, ursolic acid and rosmarinic acid using an ex vivo organotypic culture of human BC explants as an alternative model system. They showed that the explants cultured in the presence of paclitaxel plus bioactive compounds present a synergic effect, with scattered necrotic areas and a reduction of viability of neoplastic cells. Although the response to the treatments used was different in the samples from each patient, the authors suggested that there was a synergistic effect between paclitaxel with caffeic acid, ursolic acid and rosmarinic acid. In addition, it is important to study the anticancer activity or synergistic potential by evaluating natural products, a culture method and maintaining typical morphology [45].

Overall, the in vitro studies described in this review showed that the combination of conventional drugs with bioactive compounds derived from cinnamic acid improved the cytotoxic effect compared with single drugs and was able to act in important death pathways of cancer, notably apoptosis. Moreover, the combination showed a reduction in angiogenesis, invasion and migration, suggesting the inhibition of tumor metastasis. Finally, reversing resistance was observed. The main molecular mechanisms affected by the treatments are summarized in Figure 3. Other studies also revealed that cinnamic acid and its derivatives could inhibit different pathways which are essential for the proliferation of cancer cells [76] and some of these are already associated with molecular mechanisms involving apoptotic pathways, as well as anti-metastatic activity [34,77]. It is known that BC is a complex disease and cancers use different routes to escape therapy-induced cell killing and acquire drug resistance [24]. In addition, heterogeneity in individual tumor cells and cells comprising the tumor microenvironment is another challenge that promotes all modes of cancer drug resistance [78]. With advances in isolation technology and chemical synthetic capability, drug combinations have been more defined and sophisticated. So, multiple drugs with different mechanisms of action may affect a single target or disease and treat it more effectively [23,24].

This systematic review was produced through several phases. We used six databases to provide a robust, stable and comprehensive metric process in the search for publications. The identification and choice of the MeSH terms were cautious and the eligibility criteria of the selected articles, which were decided by several researchers by consensus, provides high sensitivity and specificity. The studies with in vitro methodology were selected to address different compounds and drug combinations, types of BC and different methods of analysis. A limitation of this study was the selection of articles available only in the English language. Published papers available on the preselected databases were the only papers available to be reviewed, which may have skewed the findings. In vitro studies, which met the inclusion criteria, were assessed in this review. 

Throughout this review, we highlighted those in vitro studies combining conventional drugs with bioactive compounds derived from cinnamic acid illustrates a real promise as anti-BC therapeutics options because of the excellent results of improved drug effects, increased cytotoxicity, antimigratory effect and reversing resistance (Figure 4). However, the number of recent studies is quite small and major research requirements are needed to evaluate these combinations in the following areas: (a)Other cell lines representing different types of BC and treatment-resistant BCs;(b)Other biocompounds derived from cinnamic acid;(c)Preclinical animal studies and clinical trials to confirm preclinical in vitro studies and orientate future research;(d)Exact mode of actions in different BC types.

**Figure 4 biomedicines-11-00275-f004:**
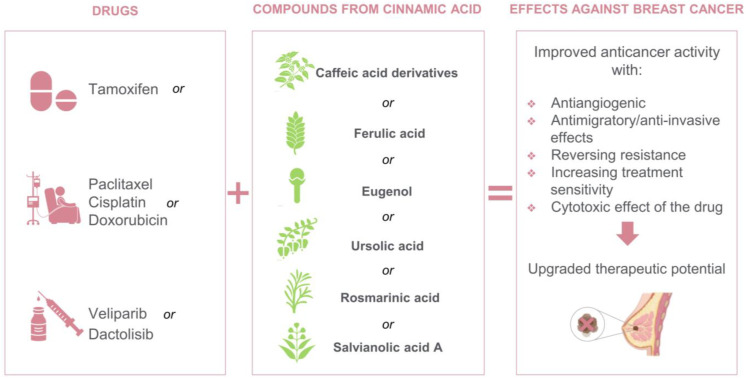
Drugs and biocompounds derived from cinnamic acid studied in vitro as combination therapy for BC.

Finally, despite all the drawbacks and limitations mentioned above, the authors firmly believe that combining conventional drugs with bioactive compounds derived from cinnamic acid could emerge as a privileged scaffold for the establishment of new treatment options for BC. A deeper understanding of the effects of combining conventional drugs with bioactive compounds derived from cinnamic acid could enable the improvement of currently used protocols to treat different BC types in the future.

## Figures and Tables

**Figure 1 biomedicines-11-00275-f001:**
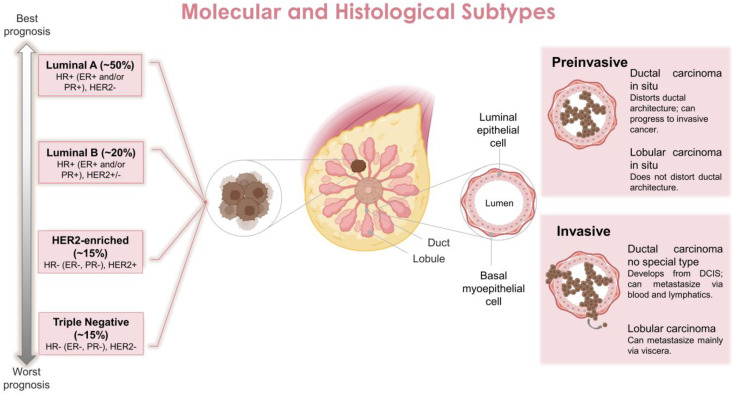
Main molecular and histological breast cancer subtypes. Abbreviations: DCIS, ductal carcinoma in situ; ER, estrogen receptor; HER2, human epidermal growth factor receptor 2; HR, hormone receptor; PR, progesterone receptor.

**Figure 2 biomedicines-11-00275-f002:**
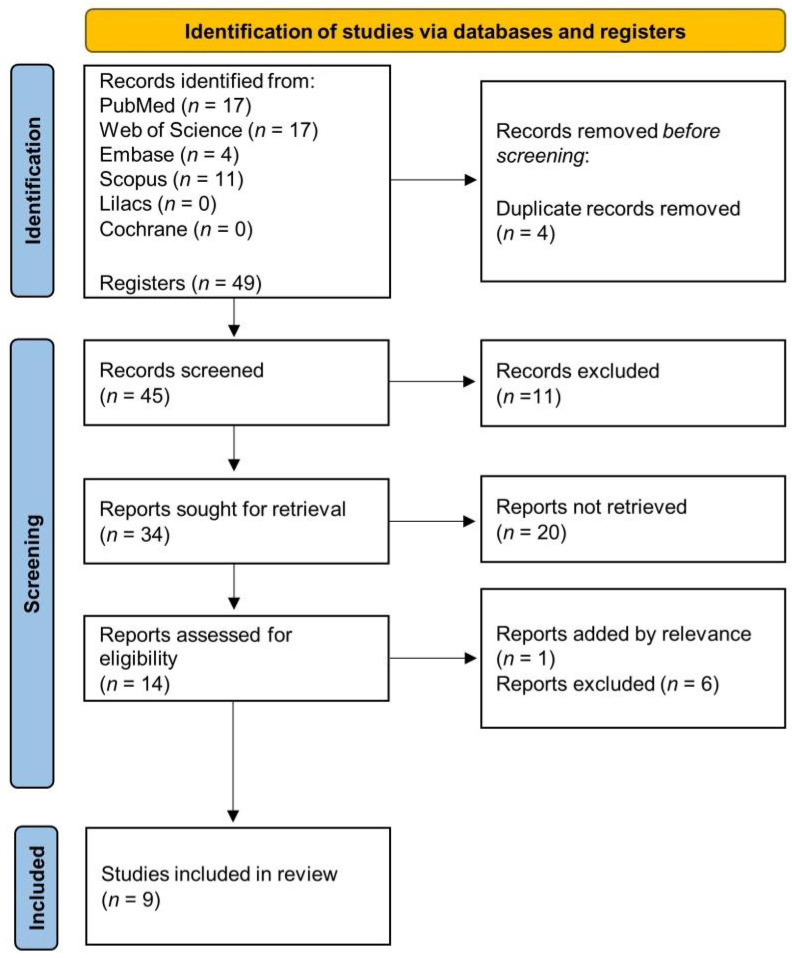
Flow chart of the article selection process.

**Figure 3 biomedicines-11-00275-f003:**
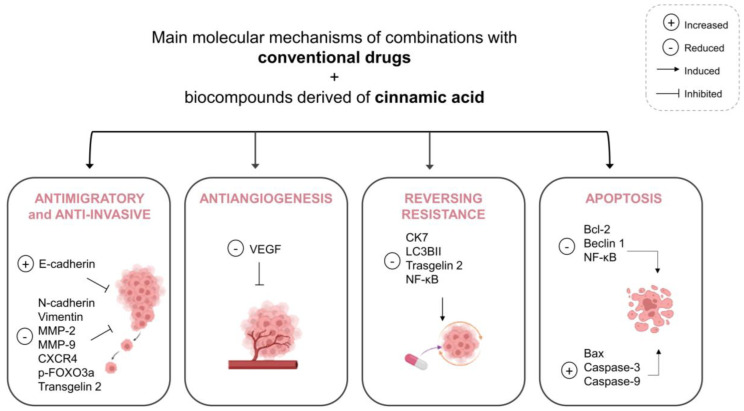
Summary of the main molecular mechanisms affected by the combination of conventional drugs and bioactive compounds derived from cinnamic acid. Abbreviations: Bax, Bcl-2-associated X protein; Bcl-2, B-cell lymphoma-2; CK7, Cytokeratin 7; CXCR4, C-X-C chemokine receptor type 4; FOXO3a, forkhead box O3; LC3BII, microtubule associated protein 1 light chain 3 beta II; MMP-9, Matrix metallopeptidase 9; MMP-2, Matrix metallopeptidase 2; NF-κB, nuclear factor kappa B; VEGF, vascular endothelial growth factor.

**Table 1 biomedicines-11-00275-t001:** Studies using conventional drug combinations with biocompounds derived from cinnamic acid against breast cancer by in vitro methodology.

References	Conventional Drug	Cinnamic Acid Biocompound	BC Cell Line	Main Analysis	Key Findings
[37]	Cisplatin	3-caffeoylquinic acid	MCF-7	In vitro growth inhibition assays and IC_50_ modulation experiments	Reduced the IC_50_ and promoted a 2.5-fold enhancement in the effect of cisplatin.
[38]	Tamoxifen	CAPE	MCF-7	Cytotoxicity assay, evaluation of drug interaction, determination of caspase-3 activity, caspase-9, Bcl-2, LC3-I and -II, beclin-1 and VEGF.	Evident cytotoxic action with lower tamoxifen and higher CAPE concentrations; increased cell death via caspase-3 e 9 and significant reduction of a factor related with angiogenesis (VEGF).
[39]	Tamoxifen	CAPE	MCF-7	Cytotoxicity assay, evaluation of drug interaction, determination of caspase-9, LC3-II, caspase-3, Bcl-2, beclin-1, VEGF, glutathione, and nitric oxide, analysis for cell death and DNA fragmentation.	Enhanced tamoxifen cytotoxicity via a multitarget approach, including weakening of autophagy, strengthening of both apoptotic and angiostatic potentials and increasing both glutathione and cellular nitric oxide levels.
[40]	Doxorubicin	Eugenol	MCF-7	Cytotoxicity assay, cell cycle and apoptosis analysis, determination of TNFα, IFNγ, FOXP3, Bax, Bcl-2, and caspase 8 genes, analysis of aromatase, EGFR, CK7, and LC3B antibodies and caspase-3, histones extraction and the determination of global H3 and H4 acetylation and activity of multidrug resistance (MDR).	Increased cytotoxic activity of doxorubicin with synergized cytotoxicity in HR+ breast cancer cells, mainly through the non-MDR pathway of histones acetylation and immunomodulation.
[41]	Paclitaxel	Salvianolic acid A	MCF-7/ PTX	Cytotoxicity assay, wound healing scratch assay, transwell invasion assay, analysis of E-cadherin, N-cadherin, Vimentin and transgelin 2.	Reversed paclitaxel resistance and inhibited invasion, migration, and growth in a dose-dependent manner.
[44]	Veliparib	Ferulic acid	MDA-MB-231 and MCF-7	Colony assay (cell survival analysis).	Increased sensitivity to the PARP inhibitor in both BC cell lines
[42]	Dactolisib	CAPE	MDA-MB-231	Cytotoxicity assay, analysis of apoptosis, surface expression of CXCR4, analysis of phospho-FOXO3a or pan-Akt antibodies and CXCR-4 and TWIST-1 genes.	Inhibited cell growth and reduced tumor metastasis.
[43]	Cisplatin	Eugenol	MDA-MB-231, MDA-MB-468 and BT-20	Cytotoxicity assay, apoptosis analysis, invasion assay, analysis of caspase 3, caspase 9, Bax, Bcl-2, MMP-2 and MMP-9, colony formation assay and sphere formation assay.	Increased cytotoxicity and pro-apoptotic effects, mediated through suppressing breast cancer stem cells self-renewal and activity.
[45]	Paclitaxel	Caffeic acid, rosmarinic acid and ursolic acid	NA	Viability assay, lactate dehydrogenase assessment, histopathological analysis, immunohistochemistry for Ki-67 expression in infiltrating ductal adenocarcinoma specimens.	A synergistic effect was observed. Promoted reduction of >40% in the population of necrotic cells with widespread necrotic areas. The response to the treatments was different in the samples from each patient.

Abbreviations; Akt, Protein kinase B; Bax, Bcl-2-associated X protein; Beclin-1, the mammalian ortholog of yeast ATG6; Bcl-2, B-cell lymphoma-2; CAPE, caffeic acid phenethyl ester; CK7, Cytokeratin 7; CXCR4, C-X-C chemokine receptor type 4; EGFR, epidermal growth factor receptor; FOXP3, forkhead box P3 protein; FOXO3a, forkhead box O3; HR+, hormone receptor positive; IC_50_, concentration that inhibited cell growth by 50%; IFNγ, Interferon gamma; Ki-67, Ki-67 protein; LC3B, microtubule associated protein 1 light chain 3 beta; LC3BII, microtubule associated protein 1 light chain 3 beta II; MCF-7/PTX, MCF-7 paclitaxel-resistant; MDR, multidrug resistance; MMP-9, Matrix metallopeptidase 9; MMP-2, Matrix metallopeptidase 2; NA, not applicable; PARP, Poly(ADP-ribose) polymerase; TNFα, Tumor Necrosis Factor alpha; TWIST-1, Twist-related protein 1; VEGF, vascular endothelial growth factor.

## Data Availability

Not applicable.

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
