# Peer review of "Combination of Conventional Drugs with Biocompounds Derived from Cinnamic Acid: A Promising Option for Breast Cancer Therapy"

_biomedicines, 2023, doi:10.3390/biomedicines11020275_

Round 1
Reviewer 1 Report
Worldwide, breast cancer (BC) is still the most frequent malignancy, representing 11.7% of all cancer cases in this population. Oncological therapy in the cases of breast cancershows a mssive expansion in the last period and knowing the tumor molecular aspects facilitates targeted therapy. Hybrid therapies that include conventional drugs and those with biological derivates represents an important step towards obtaining optimal results in the case of breast cancer patients, optimizing long-term survival.
The title and content of the article represent a topic of real interest worldwide. Combining conventional drugs with cinnamic acid derivatives can represent a hybrid therapy in breast cancer patients with the improvement of tumor response through bidirectional potentiation. The subject of the study is topical with real interest for the future.
The introduction of the article presents originality by proposing a topic with a huge academic potential..
The bibliographic data inserted along the article presents a qualitative chronology. The subject of the article represents a true scientific revolution in its field.
The material and methods section of the article presents a quantitative and qualitative exposition of the research plan, respectively a good reproducibility in order to develop other studies with this theme. I consider it necessary to develop new studies on this subject and implement them on a population scale.
The results of the article present a logical and chronological exposition outlining qualitative aspects of a quite rare disease in young population. The figures and tables keep a specific chronology throughout their exposition, presenting qualitative aspects related to the subject of the article.
The topic of the article is a real interest for the future with major importance in this field. I consider it necessary to develop new studies on this subject and implement them on a population scale. The article presents an important research point with an optimal linguistic exposition, having an exponential potential for the future.
This present article is written in a clear and concise manner. Combining conventional drugs with bioactivecompounds derived from cinnamic acid could emerge as a privileged scaffold for the establishment of new treatment options for BC
The article presents originality, with an optimal literary exposition, representing a topic of real interest for the future with objective results at the research level. The article represents a launching platform in its field and from the point of view of the characteristics it is included for publication.
Author Response
Reviewer: 1
Comments and Suggestions for Authors:
“Worldwide, breast cancer (BC) is still the most frequent malignancy, representing 11.7% of all cancer cases in this population. Oncological therapy in the cases of breast câncer shows a mssive expansion in the last period and knowing the tumor molecular aspects facilitates targeted therapy. Hybrid therapies that include conventional drugs and those with biological derivates represents an important step towards obtaining optimal results in the case of breast cancer patients, optimizing long-term survival.
The title and content of the article represent a topic of real interest worldwide. Combining conventional drugs with cinnamic acid derivatives can represent a hybrid therapy in breast cancer patients with the improvement of tumor response through bidirectional potentiation. The subject of the study is topical with real interest for the future.
The introduction of the article presents originality by proposing a topic with a huge academic potential..
The bibliographic data inserted along the article presents a qualitative chronology. The subject of the article represents a true scientific revolution in its field.
The material and methods section of the article presents a quantitative and qualitative exposition of the research plan, respectively a good reproducibility in order to develop other studies with this theme. I consider it necessary to develop new studies on this subject and implement them on a population scale.
The results of the article present a logical and chronological exposition outlining qualitative aspects of a quite rare disease in young population. The figures and tables keep a specific chronology throughout their exposition, presenting qualitative aspects related to the subject of the article.
The topic of the article is a real interest for the future with major importance in this field. I consider it necessary to develop new studies on this subject and implement them on a population scale. The article presents an important research point with an optimal linguistic exposition, having an exponential potential for the future.
This present article is written in a clear and concise manner. Combining conventional drugs with bioactivecompounds derived from cinnamic acid could emerge as a privileged scaffold for the establishment of new treatment options for BC.
The article presents originality, with an optimal literary exposition, representing a topic of real interest for the future with objective results at the research level. The article represents a launching platform in its field and from the point of view of the characteristics it is included for publication.”
Answer: Thanks for your comments on our article.
Reviewer 2 Report
Meirelles et al. describe the potential benefits of using bioactive compounds produced from cinnamic acid in combination with conventional drugs used to treat breast cancer. The systematic review is well-organized overall and stresses the use of conventional drugs in combination with bioactive components produced from cinnamic acid, which may prove to be a valuable building block for developing novel therapeutic options for various BC subtypes. Regardless of that it is an interesting systematic review on a very critical insights, the systematic review has to be revised in order to fulfill the high quality requirements of the journal.
1. The authors mentioned that they explored the use of combination drugs with bioactive compounds derived from cinnamic acid against BC in the last 10 years (from January 1, 2012, to June 29, 2022). What about from June 29, 2022, to the present? So, I highly recommend having a look at the last six months, and all updated studies should be included in this systematic review.
2. The resolution in Figs. 1, 2, and 3 is insufficient, and many details are unclear.
3. In a schematic figure, authors should summarize all of the molecular mechanisms affected by the combination of conventional drugs and bioactive compounds derived from cinnamic acid, such as FOXO3, PI3K/Akt/mTOR, CXCR4, TGF-β, NF/kB, caspases, and c-Myc.
4. Reference in Line 324 Choi et al. (2015); Line 376 Carranza-Torres et al. (2015) should be replace by its number according to the reference style of the journal
5. All abbreviations must be revised throughout the entire text, and only when they are used for the first time in full can they be used as an abbreviation for the entire word.
6. In Materials and Methods lines from 118 to 124, the authors mention the journal instructions for the authors. It doesn't make sense to me.
7. In Table 1, Reference 45, the breast cancer cell line should be mentioned.
8. I recommend putting more emphasis on the systematic review's discussion.
9. There are grammatical issues throughout the entire review, which needs to be carefully proofread.
Author Response
Manuscript “Combination of Conventional Drugs with Biocompounds Derived from Cinnamic Acid: A Promising Option for Breast Cancer Therapy”- (biomedicines-2149610).
POINT-TO-POINT REVIEWERS SUGGESTIONS
Reviewer: 2
Comments and Suggestions for Authors
“Meirelles et al. describe the potential benefits of using bioactive compounds produced from cinnamic acid in combination with conventional drugs used to treat breast cancer. The systematic review is well-organized overall and stresses the use of conventional drugs in combination with bioactive components produced from cinnamic acid, which may prove to be a valuable building block for developing novel therapeutic options for various BC subtypes. Regardless of that it is an interesting systematic review on a very critical insights, the systematic review has to be revised in order to fulfill the high quality requirements of the journal.”
Answer: Thanks for your comments on our article. All recommendations were made (Please, see text and point-to-point below).
“1. The authors mentioned that they explored the use of combination drugs with bioactive compounds derived from cinnamic acid against BC in the last 10 years (from January 1, 2012, to June 29, 2022). What about from June 29, 2022, to the present? So, I highly recommend having a look at the last six months, and all updated studies should be included in this systematic review.”
Answer: Thanks for your recommendation.
We have made a comprehensive and systematic search of relevant databases (PubMed, Embase, Web of Science, SCOPUS, Lilacs, and Cochrane library) to identify original studies published in peer-reviewed journals in English, from June 29, 2022, to the present, and the results were changed as follows:
…“ A total of 49 articles from PubMed, Embase, Web of Science, SCOPUS, Lilacs and Cochrane library were found. Four articles were excluded due to duplicates, 11 were ineligible (reviews and outside of 10 years), and further analysis of the titles and abstracts of the remaining 34 articles led to the exclusion of 20 articles (excluded based on title and abstract). In addition, 6 articles were excluded after full reading and 1 article was added from retrieved by previous search” ... (Please, see Results).
“2. The resolution in Figs. 1, 2, and 3 is insufficient, and many details are unclear.”
Answer: Thanks for your recommendation.
We have improved the resolutions of the figures as recommended. (Please, see Figs.).
“3. In a schematic figure, authors should summarize all of the molecular mechanisms affected by the combination of conventional drugs and bioactive compounds derived from cinnamic acid, such as FOXO3, PI3K/Akt/mTOR, CXCR4, TGF-β, NF/kB, caspases, and c-Myc.”
Answer: Thanks for your recommendation.
We have made the inclusion of a schematic figure that summarize the main molecular mechanisms affected by the combination of conventional drugs and bioactive compounds derived from cinnamic acid. (Please, see Fig. 3).
“4. Reference in Line 324 Choi et al. (2015); Line 376 Carranza-Torres et al. (2015) should be replace by its number according to the reference style of the journal.”
Answer: Thanks for your recommendation.
We have made the alteration and replaced the references according to the style of the journal. (Please, see text).
“5. All abbreviations must be revised throughout the entire text, and only when they are used for the first time in full can they be used as an abbreviation for the entire word.”
Answer: Thanks for your recommendation.
We have made the corrections suggested and revised all abbreviations throughout the entire text. (Please, see text).
“6. In Materials and Methods lines from 118 to 124, the authors mention the journal instructions for the authors. It doesn't make sense to me.”
Answer: Thanks for your comments and the journal instructions for the authors were removed. (Please, see Materials and Methods).
“7. In Table 1, Reference 45, the breast cancer cell line should be mentioned.”
Answer: Thanks for your recommendation.
Reference 45 used an ex vivo organotypic culture of human breast cancer explants as an alternative model system, therefore we did not include the breast cancer cell line in Table 1. However, we removed the simbol (-) and included the term “NA (not applicable)” to make the table information clearer. (Please, see Table 1).
“8. I recommend putting more emphasis on the systematic review's discussion.”
Answer: Thanks for your recommendation.
We have made the change suggested and we altered some points in the Results and Discussion section, as well as included more references to putting more emphasis on the systematic review's discussion. (Please, see Results and Discussion).
“9. There are grammatical issues throughout the entire review, which needs to be carefully proofread.”
Answer: Thanks for your recommendation.
The review text was carefully proofread and the grammatical issues were corrected. (Please, see text).
Round 2
Reviewer 2 Report
The authors have addressed all my queries. I believe the systematic review is ready for publication.